# Safety Following COVID-19 Booster Vaccine with BNT162b2 Compared to mRNA-1273 in Solid Cancer Patients Previously Vaccinated with ChAdOx1 or CoronaVac

**DOI:** 10.3390/vaccines11020356

**Published:** 2023-02-03

**Authors:** Passakorn Wanchaijiraboon, Panot Sainamthip, Nattaya Teeyapun, Sutima Luangdilok, Yong Poovorawan, Nasamon Wanlapakorn, Suebpong Tanasanvimon, Virote Sriuranpong, Thiti Susiriwatananont, Nicha Zungsontiporn, Nussara Pakvisal

**Affiliations:** 1Phrapokklao Cancer Center of Excellence, Phrapokklao Clinical Research Center, Phrapokklao Genomic Laboratories, Phrapokklao Hospital, Mueang District, Chantaburi 22000, Thailand; 2Department of Pharmacology, Faculty of Medicine, Chulalongkorn University, Bangkok 10330, Thailand; 3Division of Medical Oncology, Department of Medicine, Faculty of Medicine, Chulalongkorn University and The King Chulalongkorn Memorial Hospital, Bangkok 10330, Thailand; 4Department of Biochemistry, Faculty of Medicine, Chulalongkorn University, Bangkok 10330, Thailand; 5Center of Excellence in Clinical Virology, Department of Pediatrics, Faculty of Medicine, Chulalongkorn University and the King Chulalongkorn Memorial Hospital, Bangkok 10330, Thailand

**Keywords:** ChAdOx1, CoronaVac, BNT162b2, mRNA-1273, booster COVID-19 vaccine, safety, adverse events, cancer patients, malignancy

## Abstract

Safety data following the COVID-19 booster mRNA vaccine in solid cancer patients are scarce. We prospectively evaluated adverse events after a booster dose of the BNT162b2 vaccine as compared to the mRNA-1273 vaccine in solid malignancy patients who had previously received two doses of ChAdOx1 or heterogenous CoronaVac/ChAdOx1. Data regarding solicited and unsolicited adverse events were collected using questionnaires. The primary endpoint was the difference in incidence and severity of adverse events between BNT162b2 and mRNA-1273 vaccines. A total of 370 subjects were enrolled, including 172 (47%) and 198 (54%) patients receiving booster doses of BNT162b2 and mRNA-1273 vaccines, respectively. The overall incidence of adverse events in the two groups was comparable (BNT162b2 vs. mRNA-1273; 63% vs. 66%, *p* = 0.6). There was no significant difference in severity, and the majority of adverse events reported were classed as mild to moderate. Tenderness at the injection site was the only reaction that had a statistically higher reported incidence after the mRNA-1273 vaccine than after the BNT162b2 vaccine (56% vs. 41%, *p* = 0.003). In conclusion, a booster dose of the mRNA vaccine, either BNT162b2 or mRNA-1273, in solid cancer patients previously vaccinated with ChAdOx1 and CoronaVac appears safe, and no new safety concerns were observed.

## 1. Introduction

The coronavirus disease 2019 (COVID-19) pandemic has impacted all health care systems, including cancer care. Patients with malignancies are classified as a high-risk group because they are at increased risk of serious complications and death from COVID-19 infection [1,2,3,4]. Many different COVID-19 vaccines have been developed and have been shown to reduce the risk of SARS-CoV-2 infection and the likelihood of developing serious illness. However, new emerging variants, such as B.1.617.2 (delta) and the B.1.1.529 (omicron), which, since 2021, have spread worldwide [5,6], coupled with the waning immunity of the vaccine [7,8,9,10,11], have raised major concerns about the effectiveness of the primary series of vaccines. Several studies show that a booster dose with messenger RNA (mRNA) can enhance protection against SARS-CoV-2 variants of concern [10,12,13,14]. Currently, the Center for Disease Control (CDC) recommends a booster dose with mRNA vaccine for everyone aged 5 years and older [15].

Safety data for the mRNA vaccine booster dose in malignancy patients are limited to studies of individuals primed with two doses of the mRNA vaccine [16,17,18]. A study of two prospective cohorts of solid cancer patients showed that the mRNA vaccine booster dose was well tolerated among all participants. The most common side effect observed after the vaccination was pain at the injection site [16,17]. According to Israel’s study, cancer patients had a statistically lower incidence of headache, muscle pain, and chills than healthy controls, with no serious side effects [17]. Another nationwide survey of more than 25,000 individuals, including solid cancer patients, found adverse events following an mRNA vaccine booster dose were generally mild in severity and generally did not require medical care [18].

However, studies of cancer patients who received a primary vaccine with a viral-vectored vaccine (ChAdOx1) or inactivated virus vaccine (CoronaVac) are lacking despite the fact that many countries deployed these vaccines in the early phase of the COVID-19 pandemic. In Thailand, most cancer patients were immunized with the primary series of two doses of the ChAdOx1 vaccine or heterogenous CoronaVac/ChAdOx1 during the early phase of the vaccination program in June 2021 [19,20,21,22]. The mRNA vaccine booster dose was recommended for late 2021 and, since then, vaccine hesitancy has been seen among many cancer patients. Such hesitation is generally due to a fear of side effects, especially among those undergoing cancer treatment.

As a result, we aimed to obtain safety data for the mRNA COVID-19 vaccine booster dose in solid cancer patients previously vaccinated with two doses of ChAdOx1 or heterogenous CoronaVac/ChAdOx1. We also planned to compare the differences in adverse events between the two types of mRNA vaccine (BNT162b2 and mRNA-1273).

## 2. Materials and Methods

### 2.1. Study Design and Participants

We conducted a multi-center prospective observational cohort study at King Chulalongkorn Memorial Hospital and Phrapokklao Hospital, Thailand. We enrolled solid cancer patients who had previously been vaccinated with a primary series of two doses of the ChAdOx1 vaccine (AstraZeneca, Cambridge, UK) or heterogenous CoronaVac (Sinovac Biotech, China)/ChAdOx1 (the first dose with CoronaVac and the second dose with ChAdOx1). Enrolled patients then received a third dose using an mRNA vaccine, i.e., either BNT162b2 (Pfizer-BioNTech, New York City, NY, USA) or mRNA-1273 (Moderna, Norwood, MA, USA). The interval between the second and third dose of the vaccine was at least 3 months but not more than 6 months. The study excluded patients with a history of SARS-CoV-2 infection or a life expectancy of less than 6 months. Patient demographics, disease characteristics, and data related to cancer treatments were reviewed and recorded. The type of cancer treatment and history of concurrent corticosteroid use were determined within 4 weeks before the third vaccination.

After vaccination, the patients were asked to self-report adverse events that did not include previous symptoms experienced due to their cancer or cancer treatments using either a paper-based or electronic online questionnaire (Appendix A for at least 7 consecutive days from the day of injection, which was defined as day 0). Solicited adverse events in the questionnaire were classified into two categories. The first was local adverse effects including pain, tenderness, swelling, and erythema at the injection site. The second comprised systemic adverse effects including fever, headache, myalgia, fatigue, nausea or vomiting, arthralgia, diarrhea, back pain, dizziness, and lymphadenopathy, which is described as the palpation of a lump or mass in the axilla or neck area. Physical examination by a physician at an oncology clinic and review of all imaging performed after the vaccination were also used to determine the incidence of lymphadenopathy. Patients reported unsolicited adverse events themselves. We also gathered additional adverse events of interest including myocarditis, vaccine-induced immune thrombotic thrombocytopenia, and thrombosis by reviewing all laboratory data and medical records.

The severity of adverse events was graded according to the FDA toxicity grading scale for healthy adults and adolescent volunteers enrolled in preventive vaccine clinical trials [23]: grade 1—mild symptoms and does not interfere with activity; grade 2—moderate symptoms and some interference with activity but not requiring medical intervention; grade 3—severe symptoms and prevents daily activity or requires medical intervention; grade 4—life-threatening symptoms requiring an emergency room visit or hospitalization. The details of each grade in each adverse event were described in the questionnaire (Appendix A).

The study was performed in accordance with the principles of the Declaration of Helsinki, with all patients providing written informed consent. The study was approved by the Institutional Review Board of the Faculty of Medicine, Chulalongkorn University (No. 486/64) and the Chanthaburi Research Ethics Committee/Region 6 (CTIREC) (No. 044/64).

### 2.2. Outcomes

The primary endpoint was the incidence and severity of adverse events following the booster dose with the BNT162b2 as compared to the mRNA-1273 vaccine in solid malignancy patients previously vaccinated with ChAdOx1 or CoronaVac. We also assessed individual risk factors associated with vaccine reactogenicity and their effect on mRNA vaccine type. The secondary endpoints were the onset and duration of adverse events and the frequency with which adverse events interrupted cancer treatment.

### 2.3. Statistical Analysis

Descriptive statistics including frequencies with percentages and mean with standard deviation were used to describe demographics, disease characteristics, and data on cancer treatments. The difference in baseline characteristics and incidence and severity of adverse events between the BNT162b2 and mRNA-1273 vaccines was analyzed by the Chi-square test. We used the Mann–Whitney U test to compare the distribution of the onset and duration of adverse events between the two types of vaccine. The association between individual risk factors and developing adverse events was evaluated using univariate logistic regression, which was used to calculate the unadjusted odds ratios and corresponding 95% confidence intervals (CIs). The significant factors from the univariate analysis were included in the multivariate analysis to determine significant independent factors, and we used these factors to adjust the impact of mRNA vaccine type. *p*-values of less than 0.05 were considered statistically significant.

We used SPSS version 28.0 (IBM Corp., Armonk, NY, USA) and Stata 15 (StataCeorp LLC, College Station, TX, USA) for the statistical analyses.

## 3. Results

### 3.1. Patient Characteristics

From December 2021 to February 2022, 370 solid malignancy patients previously vaccinated with a primary series of two doses of the ChAdOx1 vaccine or heterogenous CoronaVac/ChAdOx1 were recruited for the study. Each subject then received a third dose of an mRNA vaccine. Of these, 172/370 (46.5%) and 198/370 (53.5%) were vaccinated with BNT162b2 and mRNA-1273, respectively. The BNT162b2 group contained more elderly, female, and metastatic stage cancer individuals, but less comorbidity than the mRNA-1273 group. The distribution of cancer types and treatments was also significantly different between the two groups (Table 1)**.** The mRNA-1273 group had a higher rate of concurrent steroid use because that group contained more patients who were being treated with chemotherapy, thus more individuals taking steroids for pre-medication purposes. The interval between the second and the third vaccinations was longer among the patients who were vaccinated with BNT162b2.

### 3.2. Vaccine Safety

A total of 238 out of 370 patients (64.3%) reported adverse events after receiving a booster dose of the mRNA COVID-19 vaccine. The overall incidence of any adverse events associated with BNT162b2 vaccines was comparable to that of the mRNA-1273 vaccines (62.8% vs. 65.7%, *p* = 0.57). Although the BNT162b2 group had a higher reported incidence of grade 3 severity adverse events than the mRNA-1273 group, including all adverse events (6.5% vs. 2.3%, *p* = 0.41), there was no statistically significant difference and the most common reported severity in both groups was mild to moderate. The most common local reaction was pain at the injection site, while the most common systemic reaction was myalgia in both vaccine types. Tenderness at the injection site was the only reaction that had a statistically higher incidence after the mRNA-1273 than after BNT162b2 vaccination (56.1% vs. 40.7%, *p* = 0.003), as shown in Figure 1 and Table 2. In most solicited adverse events, the median onset was 1 day after vaccination and the median duration was 2 days (Figure 2 and Figure 3). The adverse event that was reported for the longest duration was lymphadenopathy (14 days in one patient).

At the time of data cut off (31 October 2022), no serious adverse events or deaths related to the vaccine were reported. Only one patient who received the BNT162b2 vaccine needed to interrupt their cancer treatment (Ribociclib) with a cessation of only 2 days due to a grade 3 fever. There were also no adverse events of interest, including myocarditis, vaccine-induced immune thrombotic thrombocytopenia, or thrombosis, observed. 

Results from the univariate logistic regression analysis revealed significant factors associated with any adverse events, including age, sex, concurrent steroid use, and type of primary vaccine (Table 3). We incorporated these factors into the multivariate regression analysis to evaluate the association between the type of mRNA vaccine and vaccine reactogenicity. While controlling for significant independent factors, we found that the booster dose with the BNT162b2 or mRNA-1273 vaccines resulted in a similar risk of developing any adverse events (BNT162b2 vs. mRNA-1273; OR 1.22; 95% CI, 0.78–1.92; *p* = 0.38). We also investigated this association in relation to local and systemic reactions, and there was no correlation between the type of booster mRNA vaccine and vaccine reactogenicity (Table 4 and Table 5).

## 4. Discussion

To the best of our knowledge, this is the first prospective study to assess the safety of an mRNA COVID-19 booster vaccine in solid cancer patients previously vaccinated with two doses of ChAdOx1 or heterogenous CoronaVac/ChAdOx1. When we compared our findings with the safety data of the third dose of the mRNA vaccine following ChAdOx1 in a healthy population (the COV-BOOST study) [24], we found similar results as regards the pattern of reactogenicity. Local adverse events were reported more frequently than systemic adverse events, and pain at the injection site, fatigue, myalgia, and headache were common adverse events in both studies. However, the overall incidence of side effects had a higher prevalence in the healthy population [24] as compared to our population.

In cancer patients, studies of adverse events after the COVID-19 vaccine booster dose are limited to individuals primed with two doses of the mRNA vaccine [16,17,18]. A recent study in Italy that evaluated the safety of the third dose of BNT162b2 in 142 solid malignancy patients who had previously received two doses of BNT162b2 found no risk of serious adverse events, with common side effects including pain at the injection site (64.8%), fever (24.6%), arthralgia (17.6%), headache (12%), and myalgia (9.2%) [16]. Comparing the BNT162b group in Italian study with our study, ours reported a higher incidence of headache and myalgia but less local pain, fever, and arthralgia. These differences might be related to the different type of primary vaccine or the differing baseline characteristics of the studies. The patients in the Italian study were all receiving active cancer treatment and 6% had a previously documented SAR-CoV-2 infection [16]. In our study, however, 83% of patients were undergoing treatment and no patients had a history of COVID-19 infection. Moreover, there were more elderly patients and fewer females in the Italian study than in ours. Both younger age and female sex have been linked to higher vaccine reactogenicity [25,26,27].

There are no previous studies that compare adverse events following immunization for the two types of mRNA vaccines (BNT162b2 and mRNA-1273) used as the booster dose in cancer patients. There is, however, one study, involving healthy volunteers, that reported a similar incidence for any grade of local and systemic reaction among different types of booster vaccines [24]. In our study, the overall incidence and severity of adverse events were also comparable between the two mRNA vaccine types. The mRNA-1273 group had a statistically significant higher incidence of tenderness at the injection site and more moderate severity of headache. Several adverse events, including myalgia, fatigue, and back pain, were also reported more frequently, but the lack of statistical significance was likely due to the limited number of patients who experienced the events. In the mRNA-1273 group, no cancer treatment interruption due to vaccine side effects was reported, while one patient who received the BNT162b2 vaccine during treatment with Ribociclib and Letrozole developed a grade 3 fever and needed to interrupt their Ribociclib treatment for 2 days. Developing a fever during cancer treatment is a matter of concern for patients and physicians because it is a critical sign of infection. The incidence of fever associated with vaccine reactogenicity was 11.6% and 13.6% in the BNT162b2 and mRNA-1273 groups, respectively, with a median onset of 1 day after vaccination and a median duration of 1–2 days in our cohort. These findings may help oncologists to educate and reassure cancer patients that adverse events following the booster vaccine rarely affect cancer treatment.

Regarding adverse events of interest, 5 out of 370 patients (1.4%) developed lymphadenopathy after the third vaccination. This occurred after the BNT162b2 vaccine in 4 patients (2.3%) and after the mRNA-1273 vaccine in 1 patient (0.5%). All were affected in the axillary region on the same side as the injection site. Reactive lymphadenopathy, which has been reported in relation to the primary series of the mRNA COVID-19 vaccine [28,29,30,31], is an important reaction after vaccination, because it can confound tumor staging and mimic cancer progression. The incidence of reactive lymphadenopathy following a booster vaccine in malignancy patients has not yet been assessed in a large cohort study. There is a published case series describing one patient following a third dose of BNT162b2, and one case following a third dose of mRNA-1273 [32]. Oncologists and radiologists should be aware of this reaction. A booster vaccination history should be requested from each cancer patient before interpreting imaging.

As a result of the non-randomized study design, imbalances in patient demographics and disease characteristics were present between the two booster vaccine groups. Several baseline characteristics that influence vaccine reactogenicity are documented in the literature [25,26,27,33,34]. We attempted to overcome this bias by using logistic regression analysis. From the univariate analysis, the significant factors associated with any adverse events following booster immunization were age, sex, concurrent steroid use, and type of priming vaccine. We then used these factors to adjust the impact of the mRNA booster vaccine type and found no statistically significant correlation. However, the multivariate regression analysis showed that non-elderly (<65 years) and female sex were associated with a higher risk of adverse events, whereas heterogenous CoronaVac/ChAdOx1 as the priming vaccine was associated with a lower risk of developing any adverse events after booster vaccination. Younger age and female sex have previously been identified as known factors related to an increased risk of COVID-19 vaccine reactogenicity in general and in the cancer population [25,26,27,33,34]. Our results support this finding in booster immunization. No previous studies have assessed the impact of the type of the two primary vaccine doses, including both homologous and heterologous vaccine types, on booster vaccine reactogenicity. Our study is the first to establish this association.

There were a few limitations to our study. First, certain adverse events might overlap with side effects from cancer treatment because the study included patients who were receiving cancer treatment (87%). This could lead to overreporting or underreporting of adverse effects. Second, our study was not a randomized control trial, so differences in patient demographics and disease characteristics were present between the two mRNA vaccine groups. We attempted to correct this bias by using logistic regression analysis, as mentioned above.

## 5. Conclusions

A booster dose with an mRNA COVID-19 vaccine in solid cancer patients previously vaccinated with ChAdOx1 and CoronaVac appears safe, with no new safety concerns being observed in our study. The overall incidence and severity of adverse events for the BNT162b2 and mRNA-1273 vaccines were found to be comparable. The majority of adverse effects were mild to moderate in severity, with hardly any interruption of cancer treatment being reported (only one patient in our study reported this, i.e., an incidence of less than 0.5%). Our findings will help health care professionals reduce the vaccine hesitancy associated with booster mRNA COVID-19 vaccines in cancer patients.

## Figures and Tables

**Figure 1 vaccines-11-00356-f001:**
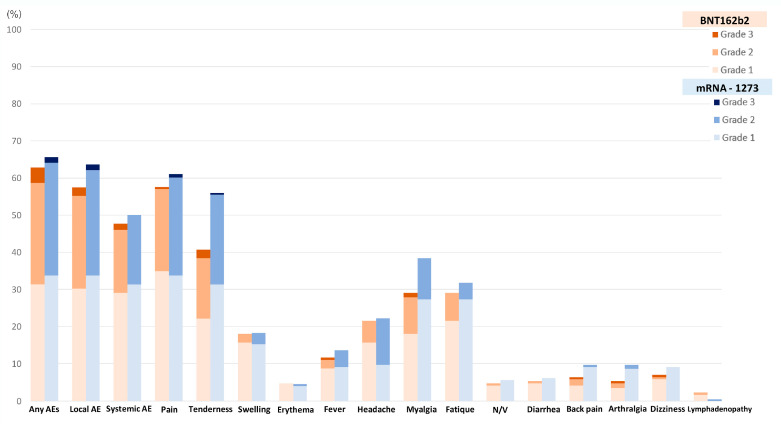
Incidence of adverse events following the booster dose with the BNT162b2 as compared to the mRNA-1273 vaccine.

**Figure 2 vaccines-11-00356-f002:**
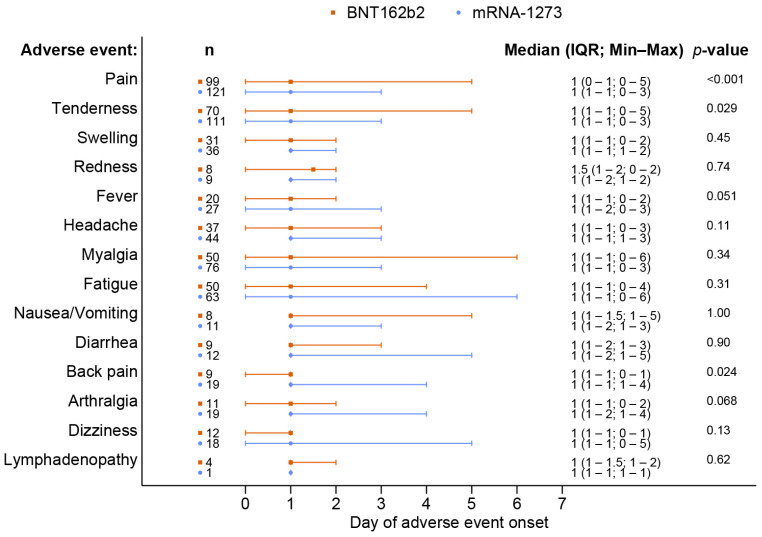
Onset of adverse events following the booster dose with the BNT162b2 as compared to the mRNA-1273 vaccine.

**Figure 3 vaccines-11-00356-f003:**
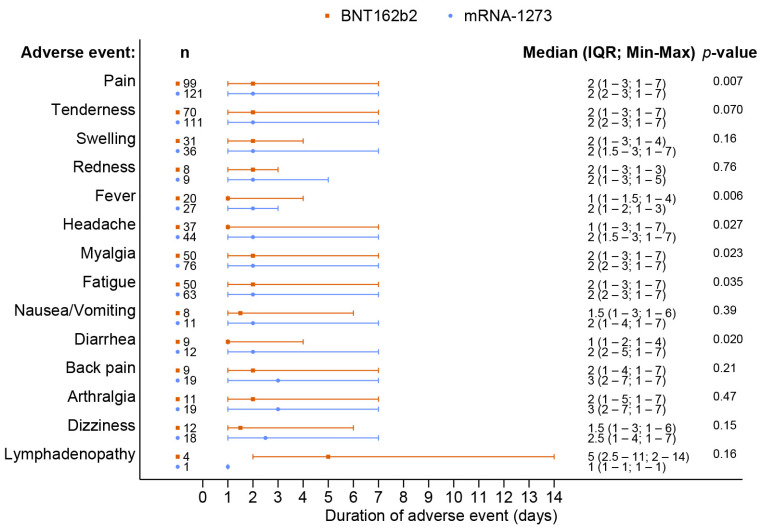
Duration of adverse events following the booster dose with the BNT162b2 as compared to the mRNA-1273 vaccine.

**Table 1 vaccines-11-00356-t001:** Patient demographics and disease characteristics.

Characteristics	All Patients *n* = 370	BNT162b2 *n* = 172	mRNA-1273 *n* = 198	*p*-Value
Age, mean +/− SD <65 years, *n* (%) ≥65 years, *n* (%)	59.61 +/− 12.3 229 (61.9) 141 (38.1)	61.12 +/− 11.9 98 (57) 74 (43)	58.30 +/− 12.5 131 (66.2) 67 (33.8)	0.027 0.070
Female, *n* (%)	213 (57.6)	108 (62.8)	105 (53)	0.058
ECOG PS, *n* (%) 0–1 2	335 (90.5) 35 (9.5)	156 (90.7) 16 (9.3)	179 (90.4) 19 (9.6)	0.923
BMI, *n* (%) <18.5 (underweight) ≥18.5	46 (12.4) 324 (87.6)	19 (11) 153 (89)	27 (13.6) 171 (86.4)	0.451
Comorbidity No Hypertension Dyslipidemia Diabetes Chronic liver disease Cerebrovascular disease Chronic obstructive pulmonary disease Chronic kidney disease Coronary heart disease Gout Autoimmune disease	175 (47.3) 128 (34.6) 81 (21.9) 66 (17.8) 19 (5.1) 12 (3.2) 7 (1.9) 6 (1.6) 5 (1.4) 5 (1.4) 3 (0.8)	66 (38.4) 67 (39) 46 (26.7) 40 (23.3) 11 (6.4) 7 (4.1) 1 (0.6) 6 (3.5) 3 (1.7) 1 (0.6) 3 (1.7)	109 (55.1) 61 (30.8) 35 (17.7) 26 (13.1) 8 (4) 5 (2.5) 6 (3) 0 2 (1) 4 (2) 0	0.001 0.100 0.041 0.012 0.311 0.400 0.088 0.008 0.542 0.231 0.059
Current disease status, *n* (%) Early Locally advanced Metastasis	111 (30) 51 (13.8) 208 (56.2)	41 (23.8) 13 (7.6) 118 (68.6)	70 (35.4) 38 (19.2) 90 (45.4)	<0.001
Cancer type, *n* (%) Breast Colorectal Lung Head neck Hepatocellular carcinoma Gastrointestinal stromal tumor Pancreaticobiliary Esophagus Gastric Prostate Bladder Melanoma Renal cell carcinoma Other	123 (33.2) 79 (21.4) 73 (19.7) 17 (4.5) 15 (4.1) 11 (3) 10 (2.7) 7 (1.9) 5 (1.4) 5 (1.4) 5 (1.4) 5 (1.4) 4 (1.1) 11 (3)	47 (27.3) 29 (16.9) 54 (31.4) 6 (3.4) 8 (4.7) 10 (5.8) 3 (1.7) 0 1 (0.6) 1 (0.6) 1 (0.6) 3 (1.7) 4 (2.3) 5 (2.9)	76 (38.4) 50 (25.3) 19 (9.6) 11 (5.5) 7 (3.5) 1 (0.5) 7 (3.5) 7 (3.5) 4 (2) 4 (2) 4 (2) 2 (1) 0 6 (3)	<0.001
Cancer treatment, *n* (%) No treatment Targeted therapy/Hormonal therapy Chemotherapy Immunotherapy Radiation	63 (17) 157 (42.4) 124 (33.5) 26 (7) 18 (4.9)	23 (13.4) 89 (51.7) 42 (24.4) 18 (10.5) 5 (2.9)	40 (20.2) 68 (34.3) 82 (41.4) 8 (4) 13 (6.6)	<0.001 0.103
Corticosteroid use, *n* (%) No Yes	261 (70.5) 109 (29.5)	133 (77.3) 39 (22.7)	128 (64.6) 70 (35.4)	0.008
Type of primary vaccine, *n* (%) ChAdOx1/ChAdOx1 CoronaVac/ChAdOx1	326 (88.1) 44 (11.9)	152 (88.4) 20 (11.6)	174 (87.9) 24 (12.1)	0.884
Interval between 2nd to 3rd vaccine, *n* (%) 3 months >3–6 months	30 (8.1) 340 (91.9)	6 (3.5) 166 (96.5)	24 (12.1) 174 (87.9)	0.002

SD—standard deviation; ECOG—Eastern Cooperative Oncology Group; PS—performance status; BMI—body mass index.

**Table 2 vaccines-11-00356-t002:** Incidence of adverse events following the booster dose with the BNT162b2 as compared to the mRNA-1273 vaccine.

Adverse Events	All Patients *n* = 370	BNT162b2 *n* = 172	mRNA-1273 *n* = 198	*p*-Value
Any AE, *n* ^a^ (%) Grade 1/2/3	238 (64.3) 121/107/10 (50.8/45/4.2)	108 (62.8) 54/47/7 (50/43.5/6.5)	130 (65.7) 67/60/3 (51.5/46.2/2.3)	0.566 0.410
Local AE, *n* ^a^ (%) Grade 1/2/3	225 (60.8) 119/99/7 (52.9/44/3.1)	99 (57.6) 52/43/4 (52.5/43.4/4.1)	126 (63.6) 67/56/3 (53.2/44.4/2.4)	0.232 0.587
Pain, *n* ^a^ (%) Grade 1/2/3	220 (59.5) 127/90/3 (57.7/40.9/1.4)	99 (57.6) 60/38/1 (60.6/38.4/1)	121 (61.1) 67/52/2 (55.4/43/1.7)	0.488 0.703
Tenderness, *n* ^a^ (%) Grade 1/2/3	181 (48.9) 100/76/5 (55.2/42/2.8)	70 (40.7) 38/28/4 (54.3/40/5.7)	111 (56.1) 62/48/1 (55.9/43.2/0.9)	0.003 0.155
Swelling, *n* ^a^ (%) Grade 1/2/3	67 (18.1) 57/10/0 (85.1/14.9/0)	31 (18) 27/4/0 (87.1/12.9/0)	36 (18.2) 30/6/0 (83.3/16.7/0)	0.968 0.666
Erythema, *n* ^a^ (%) Grade 1/2/3	17 (4.6) 16/1/0 (94.1/5.9/0)	8 (4.7) 8/0/0 (100/0/0)	9 (4.5) 8/1/0 (88.9/11.1/0)	0.961 0.331
Systemic AE, *n* ^a^ (%) Grade 1/2/3	181 (48.9) 112/66/3 (61.9/36.4/1.7)	82 (47.7) 50/29/3 (61/35.4/3.6)	99 (50) 62/37/0 (62.6/37.4/0)	0.655 0.275
Fever, *n* ^a^ (%) Grade 1/2/3	47 (12.7) 33/13/1 (70.2/27.7/2.1)	20 (11.6) 15/4/1 (75/20/5)	27 (13.6) 18/9/0 (66.7/33.3/0)	0.563 0.333
Headache, *n* ^a^ (%) Grade 1/2/3	81 (21.9) 46/35/0 (56.8/43.2/0)	37 (21.5) 27/10/0 (73/27/0)	44 (22.2) 19/25/0 (43.2/56.8/0)	0.869 0.007
Myalgia, *n* ^a^ (%) Grade 1/2/3	126 (34.1) 85/39/2 (67.5/31/1.6)	50 (29.1) 31/17/2 (62/34/4)	76 (38.4) 54/22/0 (71.1/28.9/0)	0.059 0.161
Fatigue, *n* ^a^ (%) Grade 1/2/3	113 (30.5) 91/22/0 (80.5/19.5/0)	50 (29.1) 37/13/0 (74/26/0)	63 (31.8) 54/9/0 (85.7/14.3/0)	0.567 0.118
Nausea/Vomiting, *n* ^a^ (%) Grade 1/2/3	19 (5.1) 18/1/0 (94.7/5.3/0)	8 (4.7) 7/1/0 (87.5/12.5/0)	11 (5.6) 11/0/0 (100/0/0)	0.694 0.228
Diarrhea, *n* ^a^ (%) Grade 1/2/3	21 (5.7) 20/1/0 (95.2/4.8/0)	9 (5.2) 8/1/0 (88.9/11.1/0)	12 (6.1) 12/0/0 (100/0/0)	0.731 0.237
Arthralgia, *n* ^a^ (%) Grade 1/2/3	30 (8.1) 25/4/1 (83.3/13.3/3.3)	11 (6.4) 7/3/1 (63.6/27.3/9.1)	19 (9.6) 18/1/0 (94.7/5.3/0)	0.261 0.079
Back pain, *n* ^a^ (%) Grade 1/2/3	28 (7.6) 23/4/1 (82.1/14.3/3.6)	9 (5.2) 6/2/1 (66.7/22.2/11.1)	19 (9.6) 17/2/0 (89.5/10.5/0)	0.113 0.214
Dizziness, *n* ^a^ (%) Grade 1/2/3	30 (8.1) 28/1/1 (93.3/3.3/1)	12 (7) 10/1/1 (83.3/8.3/8.3)	18 (9.1) 18/0/0 (100/0/0)	0.457 0.200
Lymphadenopathy, *n* ^a^ (%) Grade 1/2/3	5 (1.4) 3/2/0 (60/40/0)	4 (2.3) 3/1/0 75/25/0	1 (0.5) 0/1/0 0/100/0	0.130 0.171

^a^ Number of patients with adverse event; AE—adverse event.

**Table 3 vaccines-11-00356-t003:** Factors associated with any adverse events following the booster dose with the mRNA vaccine.

Factors	Univariate Analysis		Multivariate Analysis	
	Odds Ratio (95% CI)	*p*-Value	Odds Ratio (95% CI)	*p*-Value
Age * (years)				
≥65	Ref			
<65	1.97 (1.27–3.04)	0.002	1.87 (1.19–2.94)	0.007
Sex *				
Male	Ref			
Female	2.06 (1.34–3.17)	0.001	1.82 (1.16–2.88)	0.010
BMI				
≥18.5	Ref			
Underweight (<18.5)	0.69 (0.37–1.29)	0.240	NA	
Comorbidity				
No	Ref			
Yes	0.74 (0.48–1.13)	0.163	NA	
ECOG PS				
0–1	Ref			
2	0.82 (0.40–1.66)	0.575	NA	
Staging				
Non-metastasis	Ref			
Metastasis stage	0.87 (0.57–1.34)	0.541	NA	
Type of treatment				
No treatment	Ref			
Targeted/Endocrine therapy	1.52 (0.82–2.83)	0.184	NA	
Chemotherapy	0.70 (0.37–1.30)	0.258	NA	
Immunotherapy	0.92 (0.36–2.36)	0.862	NA	
Radiation				
No	Ref			
Yes	0.54 (0.21–1.39)	0.200	NA	
Concurrent steroid use *				
No	Ref			
Yes	0.54 (0.34–0.86)	0.009	0.66 (0.40–1.09)	0.112
Type of primary vaccine *				
ChAdOx1/ChAdOx1	Ref			
CoronaVac/ChAdOx1	0.46 (0.24–0.88)	0.018	0.49 (0.24–0.97)	0.042
Interval between 2nd to 3rd vaccine				
3 months	Ref			
>3 months	0.63 (0.27–1.47)	0.286	NA	
Type of mRNA vaccine *				
BNT162b2	Ref			
mRNA-1273	1.13 (0.74–1.74)	0.566	1.22 (0.78–1.92)	0.381

ECOG—Eastern Cooperative Oncology Group; PS—performance status; BMI—body mass index. * Independent variables (age, sex, concurrent steroid use, type of primary vaccine, type of mRNA vaccine) included in multivariable analysis

**Table 4 vaccines-11-00356-t004:** Factors associated with local adverse events following the booster dose with the mRNA vaccine.

Factors	Univariate Analysis		Multivariate Analysis	
	Odds Ratio (95% CI)	*p*-Value	Odds Ratio (95% CI)	*p*-Value
Age * (years)				
≥65	Ref			
<65	2.34 (1.52–3.61)	<0.001	2.05 (1.29–3.27)	0.002
Sex *				
Male	Ref			
Female	2.05 (1.34–3.14)	<0.001	1.78 (1.14–2.80)	0.011
BMI				
≥18.5	Ref			
Underweight (<18.5)	0.74 (0.40–1.38)	0.339	NA	
Comorbidity *				
No	Ref			
Yes	0.62 (0.40–0.94)	0.024	0.72 (0.44–1.17)	0.186
ECOG				
0–1	Ref			
2	0.58 (0.29–1.16)	0.122	NA	
Staging				
Non-metastasis	Ref			
Metastasis stage	0.85 (0.56–1.30)	0.454	NA	
Type of treatment				
No treatment	Ref			
Targeted/Endocrine therapy	1.60 (0.87–2.92)	0.130	NA	
Chemotherapy	0.75 (0.41–1.38)	0.357	NA	
Immunotherapy	0.96 (0.38–2.42)	0.928	NA	
Radiation				
No	Ref			
Yes	0.50 (0.19–1.29)	0.152	NA	
Concurrent steroid use *				
No	Ref			
Yes	0.52 (0.33–0.81)	0.004	0.59 (0.36–0.97)	0.042
Type of primary vaccine *				
ChAdOx1/ChAdOx1	Ref			
CoronaVac/ChAdOx1	0.55 (0.29–1.04)	0.066	0.58 (0.29–1.17)	0.129
Interval between 2nd to 3rd vaccine				
3 months	Ref			
>3 months	0.76 (0.35–1.67)	0.494	NA	
Type of mRNA vaccine *				
BNT162b2	Ref			
mRNA-1273	1.29 (0.85–1.96)	0.123	1.36 (0.86–2.13)	0.197

ECOG—Eastern Cooperative Oncology Group; PS—performance status; BMI—body mass index. * Independent variables (age, sex, comorbidity, concurrent steroid use, type of primary vaccine, type of mRNA vaccine) included in multivariable analysis.

**Table 5 vaccines-11-00356-t005:** Factors associated with systemic adverse events following the booster dose with the mRNA vaccine.

Factors	Univariate Analysis		Multivariate Analysis	
	Odds Ratio (95% CI)	*p*-Value	Odds Ratio (95% CI)	*p*-Value
Age * (years)				
≥65	Ref			
<65	2.00 (1.31–3.08)	0.001	1.94 (1.24–3.04)	0.004
Sex *				
Male	Ref			
Female	2.03 (1.33–3.09)	<0.001	1.85 (1.18–2.88)	0.007
BMI				
≥18.5	Ref			
Underweight (<18.5)	0.86 (0.46–1.60)	0.636	NA	
Comorbidity				
No	Ref			
Yes	0.86 (0.57–1.30)	0.480	NA	
ECOG				
0–1	Ref			
2	0.67 (0.29–1.16)	0.120	NA	
Staging				
Non-metastasis	Ref			
Metastasis stage	0.85 (0.33–1.36)	0.271	NA	
Type of treatment				
No treatment	Ref			
Targeted/Endocrine therapy	1.24 (0.69–2.22)	0.479	NA	
Chemotherapy	0.66 (0.36–1.21)	0.174	NA	
Immunotherapy	0.71 (0.28–1.79)	0.467	NA	
Radiation *				
No	Ref			
Yes	0.28 (0.09–0.86)	0.028	0.29 (0.09–0.92)	0.045
Concurrent steroid use *				
No	Ref			
Yes	0.68 (0.43–1.07)	0.096	0.88 (0.54–1.45)	0.631
Type of primary vaccine *				
ChAdOx1/ChAdOx1	Ref			
CoronaVac/ChAdOx1	0.43 (0.21–0.85)	0.015	0.41 (0.20–0.86)	0.018
Interval between 2nd to 3rd vaccine				
3 months	Ref			
>3 months	0.83 (0.39–1.74)	0.614	NA	
Type of mRNA vaccine *				
BNT162b2	Ref			
mRNA-1273	1.10 (0.73–1.65)	0.655	1.16 (0.75–1.80)	0.501

ECOG—Eastern Cooperative Oncology Group; PS—performance status; BMI—body mass index. * Independent variables (age, sex, radiation, concurrent steroid use, type of primary vaccine, type of mRNA vaccine) included in multivariable analysis.

## Data Availability

Data are available upon reasonable request to the corresponding author.

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
