# Peer review of "Safety Following COVID-19 Booster Vaccine with BNT162b2 Compared to mRNA-1273 in Solid Cancer Patients Previously Vaccinated with ChAdOx1 or CoronaVac"

_vaccines, 2023, doi:10.3390/vaccines11020356_

Round 1

Reviewer 1 Report

The manuscript is well written, interesting and the conclusions are supported by the results.

I have few minor questions:

-how was thw number of volunteers to be recruited determined? did the authors performed a sample size calculation before starting the enrollment?

-the author states that there were significant differences (lines 118-122) in the composition of the two groups recieving the BNT162b2 or the  mRNA-1273 vaccine. How were the patients assigned to the two different groups? 

Author Response

Response to Reviewer 1 comment

The manuscript is well written, interesting and the conclusions are supported by the results.

Response: We thank Reviewer 1 for your compliments.

I have few minor questions:

-how was the number of volunteers to be recruited determined? did the authors performed a sample size calculation before starting the enrollment?

Response: Thank you for your comments. To be honest, we did not calculate the sample size before starting the enrollment because initially we would like to report the outcome in a real-life setting and we had limited resources especially a shortage of mRNA vaccine at the time of enrollment. However, we conducted a multi-center study at King Chulalongkorn Memorial Hospital and Phrapokklao Hospital, Thailand, to increase the number of participants as much as possible during the study period.

-the author states that there were significant differences (lines 118-122) in the composition of the two groups recieving the BNT162b2 or the mRNA-1273 vaccine. How were the patients assigned to the two different groups? 

Response: Thank you for your comment. Due to the non-randomized study design, the participants received the BNT162b2 or the mRNA-1273, depending on the available vaccine type at the time of vaccination and patients’ preference. We realize that the differences in patient demographics and disease characteristics that occurred between two types of mRNA vaccine groups might lead to a bias, so we attempted to correct this bias by using the logistic regression analysis, as mentioned in the methodology (line 117-121, page 3) and discussion parts (line 269-272, page 16).

Reviewer 2 Report

The paper "Safety following COVID-19 booster vaccine with BNT162b2 compared to mRNA-1273 in solid cancer patients previously vaccinated with ChAdOx1 or CoronaVac" presents the results of a non randomized unicenter prospective study, that aims to analyze adverse events after a COVID-19 booster mRNA vaccine. The study has clear objectives, the methodology is appropriated and conclusions have beem supported by results. The limitations have been discussed.  

Author Response

Thank you very much for your compliments. We hope our findings are useful in providing cancer patients with knowledge that can decrease vaccine hesitancy with the booster mRNA COVID-19 vaccine.

Reviewer 3 Report

The language and writing are clear and of high quality, and the manuscript proves its important contribution to the objectives of scientific research. There is no meaningful gap in the literature cited. To sum up, it can be said that the right type of participant correctly supports the manuscript.

Author Response

(The authors gave the same response as above.)

Reviewer 4 Report

*  The introduction doesn't give credit for this kind of subject. Please reformulate it.

* The authors don't give much information related to patients, especially their chronic diseases. The selected variables of interest are not suitable to test the hypothesis.
* 370 patients is not sufficient to provide a decisive answer. The authors have a clear question but their data are not suitable to provide an answer to their research question.
* Quality of figures is so important too. Please provide some high-resolution figures. Some figures have a poor resolution.

Author Response

Response to Reviewer 4 comment

We thank reviewer 4 for all constructive criticisms. We used a paid editing service of MDPI to undergo English revisions on our revised manuscript (as shown in markup from tract change) as per your suggestion.

*  The introduction doesn't give credit for this kind of subject. Please reformulate it.

Reply: Thank you for your comment. We aggressively reviewed the previous literature about the safety data for the booster dose of the mRNA vaccine in solid malignancy patients. It is limited to studies of those who had been primed with two doses of the mRNA vaccine. However, we added more detail about it in the introduction part (line 48-64, page 2), following your suggestion. Study of patients who received a primary vaccine with viral-vectored vaccine (ChAdOx1) or inactivated virus vaccine (CoronaVac) is lacking despite the fact that many countries including Thailand deployed these vaccines in the early phase of the COVID-19 pandemic. To our knowledge, our study is the first prospective study to report the adverse events following immunization of the mRNA vaccines (BNT162b2 and mRNA-1273) as the booster dose in cancer patients who received a primary vaccine with ChAdOx1 or CoronaVac.

* The authors don't give much information related to patients, especially their chronic diseases. The selected variables of interest are not suitable to test the hypothesis.

Reply: Thank you for your comment. We reported all significant patient demographics and disease characteristics, including age, sex, BMI, comorbidity (chronic disease), current cancer status, cancer type, cancer treatment, concurrent corticosteroid use, and the interval between the second and third vaccination as shown in Table 1 (page 4-5). We know from the previous literature that these factors might influence our primary endpoint, the incidence of vaccine-related adverse events. Furthermore, we used logistic regression analysis to evaluate the association between individual factors and the risk of developing adverse events. The significant factors from the univariate analysis were included in the multivariate analysis to determine significant independent factors, and we used these factors to adjust the impact of mRNA vaccine type, as mentioned in the methodology (line 117-121, page 3). According to the result in Table 4, we found that comorbidity is a significant factor associated with local reactions in univariate analysis but loses its statistical significance in multivariate analysis. We added the result of the odd ratio (95% CI) and P-value of comorbidity in Table 4 (page 12). We apologize for forgetting to fill out this information and appreciate your suggestion.

* 370 patients is not sufficient to provide a decisive answer. The authors have a clear question but their data are not suitable to provide an answer to their research question.

Reply: Thank you for your comment. We conducted a multi-center study at King Chulalongkorn Memorial Hospital and Phrapokklao Hospital, Thailand, to increase the number of participants as much as possible during the study period. However, we had limited resources, especially a shortage of mRNA vaccine at the time of enrollment. Even though our study was an observational one, we collected and monitored the events systematically as a prospective study to minimize missing important data and increase the validity by using logistic regression analysis to adjust the independent risk factors. At this moment, we would like to report our findings in a real-life setting to reduce vaccine hesitancy in the booster mRNA vaccines among cancer patients.

* Quality of figures is so important too. Please provide some high-resolution figures. Some figures have a poor resolution.

Reply: Thank you for your important suggestion. We edited all figures (page 6,9,10) in higher resolution as shown in the revised manuscript.

Round 2

Reviewer 4 Report

Accept
